



# Structural and Seismic Monitoring of a Monumental Building:
# the Case Study of the Royal Castle of Racconigi

Gianni Niccolini[1], Amedeo Manuello[1], Elena Marchis[2], Alberto Carpinteri[1]

[1]Politecnico di Torino, Department of Structural, Geotechnical and Building Engineering,

Corso Duca degli Abruzzi 24, 10129 Torino, Italy

[2]Politecnico di Torino, Department of Architecture and Design,

Corso Duca degli Abruzzi 24, 10129 Torino, Italy

*Correspondence to*: Gianni Niccolini (gianni.niccolini@polito.it)

**Abstract.** The stability of an arch as a structural element in the thermal bath of King Carlo Alberto in the Royal Castle of Racconigi (on the UNESCO World Heritage List since 1997) was assessed by the Acoustic Emission (AE) monitoring technique with application of classical inversion methods to recorded AE data. First, damage source location by means of triangulation techniques and signal frequency analysis were carried out. Then, the recently introduced method of natural time analysis was preliminarily applied to the AE time series in order to reveal possible entrance point to a critical state of the
monitored structural element. Finally, possible influence of the local seismic and micro-seismic activity on the stability of the monitored structure was investigated. The criterion to select relevant earthquakes was based on the estimation of the size of earthquake preparation zones. The presented results suggest the use of AE technique as a tool for detecting both ongoing structural damage processes and micro-seismic activity during preparation stages of seismic events.

**1 Introduction**

Fracture in heterogeneous materials is a complex phenomenon which involves a wide range of time, space and magnitude scales, from microcracking to earthquake ruptures, including structural failures (Omori, 1894; Richter, 1958; Kanamori and



Anderson, 1975; Aki, 1981). Thus, acoustic emission (AE) monitoring during loading experiments give an insight into the evolution of microcrack networks in laboratory experiments and possibly a tool for understanding the occurrence of fractures at larger scales (Mogi, 1962). Since the last decades, this approach has provided the opportunity to develop universal scaling laws reflecting the scale-invariance and the self-similarity of fracture processes from the laboratory to the fault scale in time,

space and magnitude domains (Turcotte, 1997; Bonnet et al. 2001; Bak et al., 2002; Tosi et al., 2004; Corral, 2006; Davidsen et al., 2007; Kun et al., 2008). These studies should hopefully contribute to solving the main problems of earthquake prediction and the remaining life assessment of structural elements. In particular, the latter is a crucial issue for researchers involved in restoration projects of historic monuments with damaged and cracked structural elements, which can benefit from the use of nondestructive monitoring techniques for the structural integrity assessment (Carpinteri et al., 2011; Schiavi

et al., 2011; Lacidogna et al., 2015a). AE monitoring, as it provides information on the internal state of a material without altering state of conservation of statues, monuments and fine artworks, seems to be suitable for this kind of structural monitoring.

A relevant case study is here illustrated by the Racconigi Castle (origin dating back to the XIII century), which is located in the province of Cuneo and represents one of the most important monuments in northwestern Italy. The castle, with its

bearing walls decorated by frescoes, represents an extraordinary benchmark for the definition of conservation methods exploiting new technologies (Lacidogna et al., 2011; Niccolini et al. 2014). This paper presents the results of AE monitoring of an arch in the thermal bath of King Carlo Alberto, located in the ground floor of the Racconigi Castle. The wing of the castle containing this room is currently being restored.

On the other hand, besides the disruptive power of strong earthquakes, Italian historic buildings and monuments suffer the

action of small and intermediate earthquakes whose effects, though not immediately or clearly visible, eventually result in increased vulnerability to stronger earthquakes with catastrophic human and economic costs.  In this framework, over the recent years there has been an increasing interest in AE monitoring related to environmental phenomena. Several case histories in the Italian territory and previous studies support the hypothesis that an increased AE activity may be signature of crustal stresses redistribution in a large zone during the preparation of a seismic event (Gregori et al., 2004; Gregori et al.,

2005; Carpinteri et al., 2007; Niccolini et al., 2011). According to previous research studies performed by Dobrovolsky et al. (Dobrovolsky et al., 1979), it can be assumed that the preparation zone is a circle with the centre in the epicenter of the impending earthquake. The radius $r$ of the circle, called 'strain radius', is given by the relationship $r = 10^{0.433M + 0.6}$, where $M$ is the earthquake magnitude and $r$ is expressed in km. All the seismic precursors, including AE, are expected to fall into this circle.




## 2 The monitoring site

### 2.1 The Castle of Racconigi: an ancient Savoy Residence

The castle of Racconigi, which has medieval origins, is the southernmost of the Savoy Residences and the farthest from the city of Turin (Fig. 1). The original structure was made up of four towers arranged around a central courtyard comparable

5    with the current castle in nearby Fossano. Over the centuries, the structure of medieval castle was transformed into a country residence of the princes of Carignano. After becoming King of Sardinia, Carlo Alberto of Savoia Carignano transformed the Castle into a royal residence.

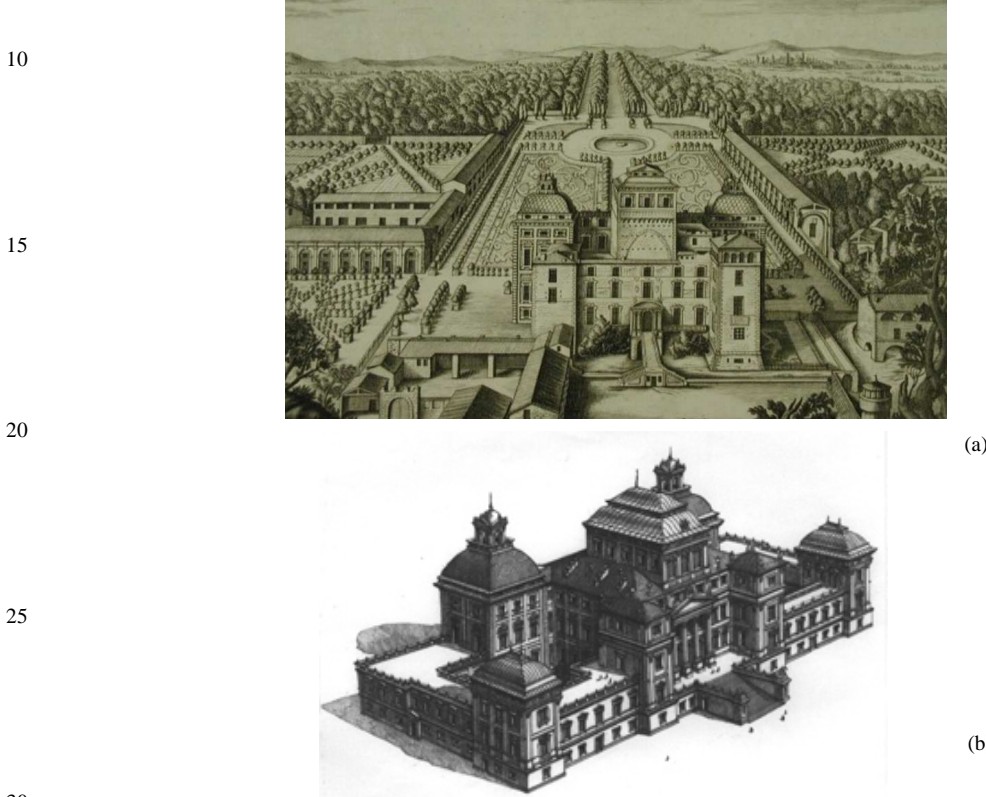

(a)

(b)

**Fig. 1: The Castle of Racconigi. Copper engraving by Bartolomeo Giuseppe Tasnière (1712) Collection Simeom (a). Axonometric view of the Castle in the contemporary configuration (b).**





Active personalities in the field of architecture contributed to the transformation of the ancient fortress into a pleasure residence. The architect Guarino Guarini (1624-1683) erected the current central section (Salone d'Ercole) where the court was, adding a pagoda-like roof, and replaced the two northern towers by two pavilions dome roof and square plan, provided with white marble lanterns. In the second half of the eighteenth century, further works in neoclassical style was due to

Giovanni Battista Borra (1713-1770), who realized the staircase towards the village and the neo-classical facade.

The moment of increased activity and restoration occurred between 1831 and 1848 during the reign of Carlo Alberto, who further enlarged and embellished the castle to represent the splendor of the newly acquired reign of Sardinia. Carlo Alberto commissioned the extension of this residence to Ernesto Melano (1784-1867) and its decoration to Pelagio Pelagi (1775-1860). Pelagi completely redesigned the castle, with the exception of the east wing which houses the Chinese apartments that

due to their preciousness and importance were saved by interventions of adjustment. In particular, Pelagi planned the complex of baths made on the Roman thermal bath style and inspired by the "balnea" of the Pompeian villas.

The complex, located on the ground floor of the west wing under the private apartment of the king and directly accessible through an internal staircase, was formed by a series of rooms with walls and floors covered with marble slabs, ceilings painted with grotesques, sumptuous baths carved in white marble blocks, brass tap the form of a swan neck. Vitale Sala

(1803 -1835) painted the vaulted ceiling, depicting the birth of Venus and the fight against animal and sea monsters.


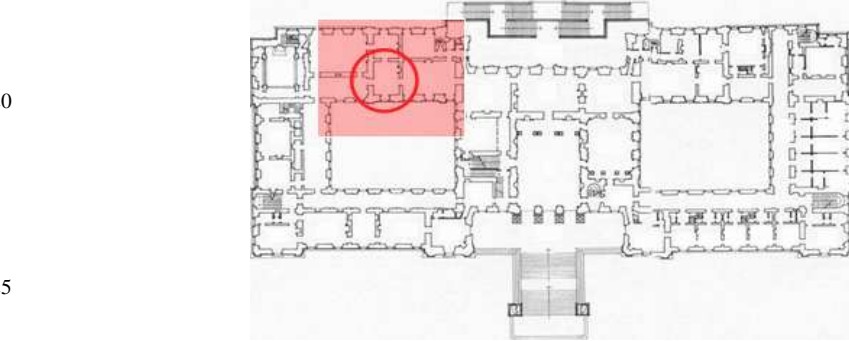


**Fig. 2: Plan of the 1 floor: localization of the monitored bearing structure inside the thermal bath.**




In the castle of Racconigi the last King of Italy Umberto II was born in 1904. Having received the castle as a wedding present in 1930, he proceeded to install in it the family gallery of some 3,000 paintings and historical documents regarding the Shroud of Turin.

During the twentieth century the castle underwent new adjustments and it was used by Savoia descendants until 1980, when it was acquired by the Italian State. The building was subjected to a restoration project, with a careful recovery of the vast park, the Royal Greenhouses and the Margaria, together with a cataloguing and restoration program of the works and furnishings held within the residence.

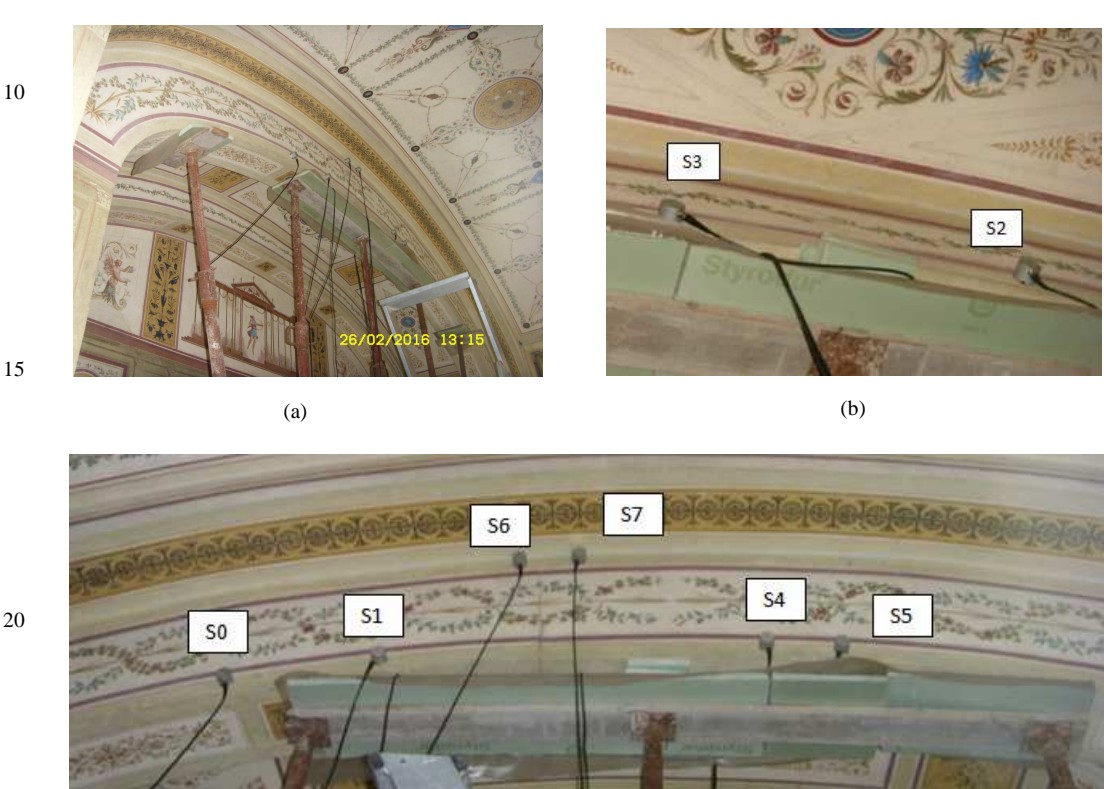

Fig. 3: The monitored arch (a) and the transducers positions (b) and (c)



### 3 Damage localization in the arch by AE Technique

The investigation of the arch structure (a masonry element of about 4 m of span) conditions, including presence and severity of cracks, is necessary to assess its performance and, possibly, to plan the restoration procedure. Damage localization in the arch of castle's thermal bath has been carried out by the AE technique. Among all structural elements, arches and vaults made of stone or brick, be they bearing or not, are the most prone to degradation and stress caused by seismic events, changes in acting loads and foundation sinking, which cause the structure to lose its original mechanical properties. Because these elements are of great historic and architectural value, they need to be consolidated in a non-invasive, compatible and consistent way with regard to their special features. The examined architectural element, which is currently supported by a steel frame structure, exhibits a relevant crack pattern. The propagation of one visible macrocrack has been investigated by an array of eight piezoelectric transducers fixed on the arch surface as shown in Fig.3. The AE transducers have been connected to a 8-channel acquisition system AEmission® which implements algorithms for automatic analysis of AE signal parameters, i.e., arrival time, duration, amplitude and counts number (total number of signal threshold crossings). The stored AE parameters can be wireless transmitted to a receiver allowing long-distance remote and real-time monitoring.

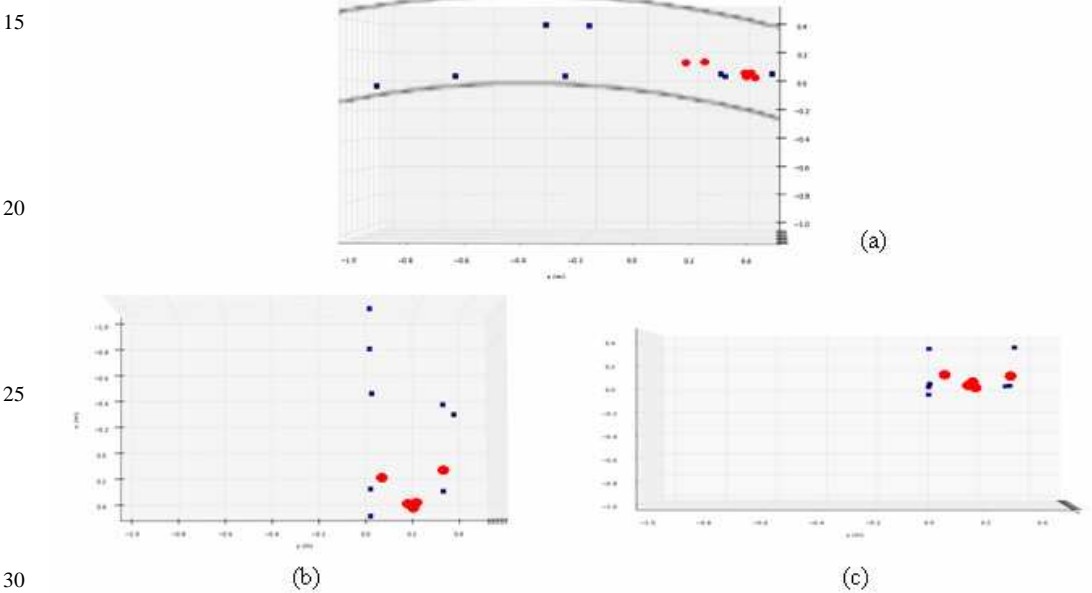

**Figure 4: 3-D map showing the positions of transducers (blue squares) and localized AE sources (red dots): front (a), upper (b) and side view (c).**





After setting the signal detection threshold to 1.5 mV in order to filter out background noise, a one-month monitoring period started. In order to suppress possible voltage spikes, acquired AE signals with duration < 3 μs and counts number < 3 have been filtered out. The spatial identification of arch's damaged zones has been performed by applying triangulation equations to the received AE signals in order to localize the AE sources as active and propagating crack tips (Shah and Li, 1994;

Shiotani et al., 1994; Grosse et al., 1997; Guarino et al., 1998; Ohtsu et al., 1998; Colombo et al. 2003; Turcotte et al., 2003; Aggelis et al., 2013). The 3D diagrams shown in Fig. 4 suggest that the arch experiences damage on one side, despite the use of reinforcing elements. In particular, the increased AE rate marked by a vertical dashed line in the top diagram of Fig. 5 can be regarded as a signature of unstable damage accumulation.

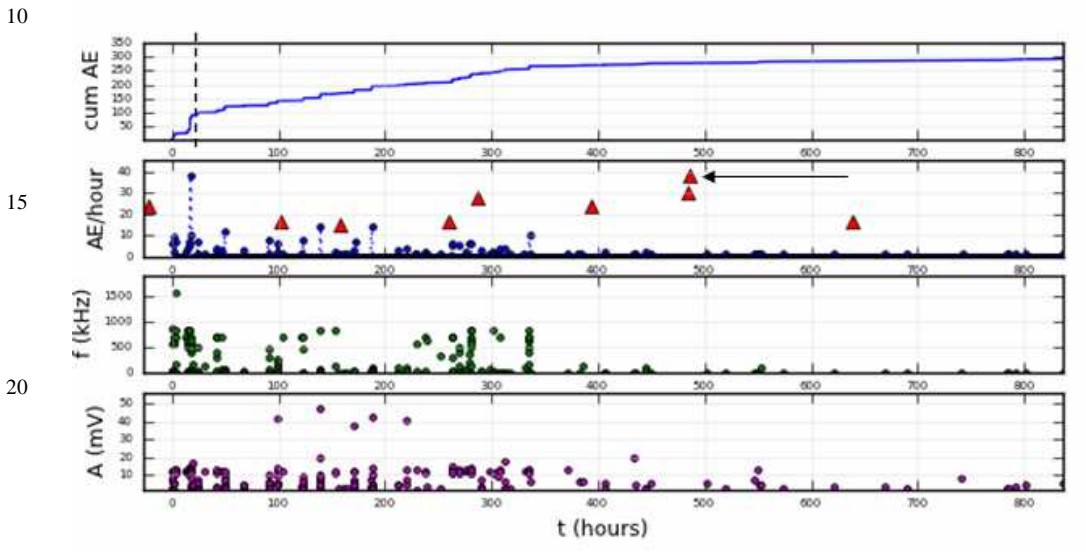



**Figure 5: From top to bottom: accumulated number of AE signals ("hits"): the dashed line indicates a critical point revealed by the natural time analysis; AE count rate and sequence of nearby earthquakes marked by red triangles (the arrow points the strongest one); time series of signal frequencies and amplitudes.**




## 4 Analysis of the arch stability and influence of nearby seismicity

In the frame of critical phenomena (Bak et al., 1989; Stanley, 1999), the fracture process is viewed as a critical state of a dynamical system and the problem of early detection of fracture precursors in structural elements along before the final collapse is transformed in the investigation of indicators revealing the entrance to a "critical state".

Recently, a worth of mentioning approach to identify when a complex system enters a critical state has been developed, based on the time-series analysis of N events read in a new time domain, termed natural time $\chi$ (Varotsos et al., 2001; Varotsos et al., 2011a; Varotsos et al., 2011b), where the time stamping is ignored and only the natural time, $\chi_k = k/N$, as a normalized order of occurrence of the $k$-th event, and the energy $Q_k$ are preserved. In natural time analysis the evolution of the pair $(\chi_k, Q_k)$ is considered, by introducing the normalized power spectrum $\Pi(\omega) \equiv |\Phi(\omega)|^2$, defined by $\Phi(\omega) = \Sigma_{i=1}^{N} p_k$

$\exp(i\omega k)$, where $\omega$ stands for the angular natural frequency and $p_k = Q_k / \Sigma_{i=1}^{N} Q_i$ is the normalized energy of the $k$-th event. It was found that all the moments of the distribution of the $p_k$ can be estimated from the Taylor expansion $\Pi(\omega) = 1 - \kappa_1 \omega^2 + \kappa_2 \omega^4 + \dots$, where the values of the coefficient $\kappa_1$, which is just the variance of natural time $\chi$, i.e., $\kappa_1 = \Sigma_{k=1}^{N} p_k \chi_k^2 - (\Sigma_{k=1}^{N} p_k \chi_k)^2 \equiv <\chi^2> - <\chi>^2$, are useful in identifying the approach of a dynamical system to a critical state. The variance $\kappa_1$ varies when a new AE event ("hit") occurs, as $(\chi_k, p_k)$ are rescaled as natural time $\chi_k$ changes from $k/N$ to $k/(N+1)$ and $p_k$ changes

to $Q_k / \Sigma_{i=1}^{N+1} Q_i$. Thus, the evolution hit by hit of $\kappa_1$ is shown along with that of the entropy $S$, which in the natural time domain is defined as $S = <\chi \ln\chi> - <\chi> \ln <\chi>$, where $<\chi \ln\chi> = \Sigma_{k=1}^{N} p_k \chi_k \ln \chi_k$.

It has been successfully shown for a variety of a variety of dynamical systems that entering the critical state occurs when the variance $\kappa_1$ converges to 0.07 (Varotsos et al., 2001; Varotsos et al., 2011a; Varotsos et al., 2011b), even if a theoretical derivation of the general validity of the $\kappa_1 = 0.07$ condition for criticality still remains an open issue. Two criteria have been

defined to identify the entrance of a system to true critical state: 1) the parameter $\kappa_1$ must approach the value 0.07 "by descending from above"; 2) the entropy $S$ must be lower than the entropy of uniform noise, $S_u = 0.0966$, when $\kappa_1$ coincides to 0.07.

In the present study the damage evolution of a structural element is investigated by properly analyzing the AE events time series following two approaches and comparing the results: the first is the analysis of AE time series in terms of natural time,

in which the evolution of variance $\kappa_1$ and entropy $S$ of the natural time series $\{ \chi_k \}$, where the aforementioned energy $Q_k$ is proportional to the seismic moment associated with AE event amplitude $A_k$, $Q_k = A_k^{1.5}$ (Kanamori and Anderson, 1975; Turcotte, 1997), while the second is the analysis of evolving AE signal frequencies over the monitoring time (Gregori et al., 2004; Gregori et al., 2005; Schiavi et al., 2011).

First, the evolution of natural time variance $\kappa_1$ and entropy $S$ as functions of the accumulated number $N$ of hits, i.e., as they

change with the addition of every new hit, is plotted in Fig. 6. Thus, it is possible to easily reveal the possible entrance point to "critical state", corresponding to the fulfillment of criticality conditions 1) and 2) (Vallianatos et al., 2013; Hloupis et al., 2015; Hloupis et al., 2016). It is worth noting that the criticality initiation point (marked with a vertical line at the $N = 104$ hit





number in Fig. 6) corresponds exactly to the abrupt jump in the AE rate highlighted in Fig. 5 and amounting to about 100 hits. This result, though obtained from a relatively small data sample, apparently confirms the potential of the AE natural time analysis to reveal the onset of criticality in fracture systems.

Second, the AE signal frequency analysis has been correlated with the nearby seismicity within a radius of 100 km from the monitoring site (see Fig. 5 with the earthquake time series marked by red triangles). In particular, it has been found that AE activity vanishes at the end of a seismic sequence culminating in a 3.0 magnitude earthquake (pointed by the black arrow), suggesting that part of the detected AE activity might be rather due to diffused microseismic activity falling in the preparation zone of the considered earthquake (strain radius > epicentral distance from the monitoring site) according to the criterion proposed by Dobrovolsky et al. (1979) (see Table 1 and Fig. 7).

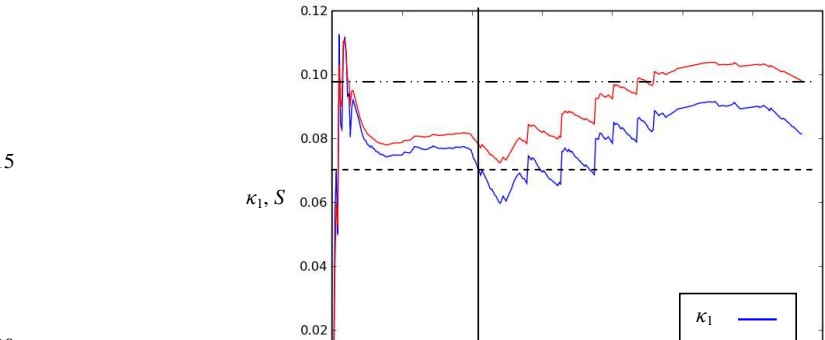

**Figure 6: Evolution of natural time quantities $\kappa_1$ (blue line) and $S$ (red line) as function of accumulated AE hit number $N$. The horizontal dashed line indicates the value $\kappa_1 = 0\ 0.07$, while the vertical line indicates the corresponding criticality initiation hit ($N = 104$). The upper dash-dot line indicates the entropy limit $S_u = 0.0966$.**

| Time Origin (UTC) | Time Delta(h) | Lat (deg) | Long (deg) | Depth (km) | Dist. (km) | Magnitude | Radius (km) |
|---|---|---|---|---|---|---|---|
| 2015-12-10 16:31:56.990 | -22.46 | 44.561 | 7.161 | 11.2 | 46.8 | 1.9 | 26.5 |




| | | | | | | |
|---|---|---|---|---|---|---|
| 2015-12-15 11:21:44.070 | 102.73 | 45.087 | 7.163 | 7.9 | 53.6 | 1.3 | 14.5 |
| 2015-12-18 05:36:07.740 | 158.6 | 44.549 | 6.775 | 5.6 | 75.3 | 1.2 | 13.2 |
| 2015-12-22 07:10:41.190 | 259.68 | 44.93 | 6.874 | 10.4 | 65.7 | 1.3 | 14.5 |
| 2015-12-23 13:45:30.560 | 286.75 | 44.651 | 6.84 | 6.8 | 67.3 | 2.2 | 35.7 |
| 2015-12-28 01:11:29.380 | 394.18 | 44.45 | 7.289 | 11.8 | 46.8 | 1.9 | 26.5 |
| 2015-12-31 19:16:48.900 | 484.26 | 44.548 | 6.756 | 9.1 | 76.8 | 2.4 | 43.6 |
| **2015-12-31 20:42:01.000** | **485.7** | **44.765** | **6.76** | **9.4** | **71** | **3.0** | **79.2** |
| 2016-01-07 05:41:23.000 | 638.68 | 45.07 | 6.57 | 10 | 93.2 | 1.3 | 14. 5 |

**Table 1: List of nearby earthquakes occurred during the AE monitoring; the event with preparation zone embedding the monitoring site is written in bold.**

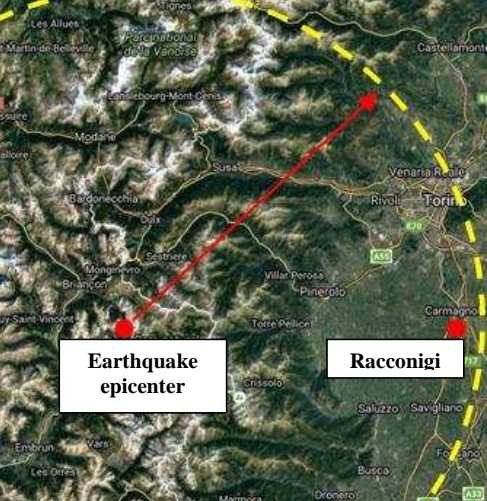

**Figure 7: Map showing the epicenter of the magnitude 3 earthquake occurred on 2015-12-31 and the related**
15 **preparation zone embedding the monitoring site in Racconigi.**





As cracking is a multi-scale phenomenon in the Earth's crust, the frequencies of AE waves related to micro-seismic activity are spread over a broad spectrum. At the earlier stages of the preparation of a seismic event, mainly micro-cracks will be present and active and therefore high-frequency AE (MHz), while finally large cracks and lower frequencies will prevail, reaching also the audible field (Hz) during the earthquake occurrence (Gregori et al., 2004; Gregori et al., 2005; Carpinteri,

5    2015).

Hence, we have subjected the distribution of the AE signal frequencies (calculated by dividing the counts number to the signal duration) to a statistical analysis, by partitioning the time window preceding the considered seismic event into three sub-intervals (0-80h; 80h-285h; 285h-485h) roughly containing the same number of AE events. Comparing the plots of the corresponding distributions (Fig. 8), a progressive reduction of the highest frequencies, i.e., between 400 and 800 kHz, is

10    observable as the seismic event was approached. The frequency decay over the time emerges also from the decreasing trend of the distributions' mean values, which are 224, 184 and 132 kHz.

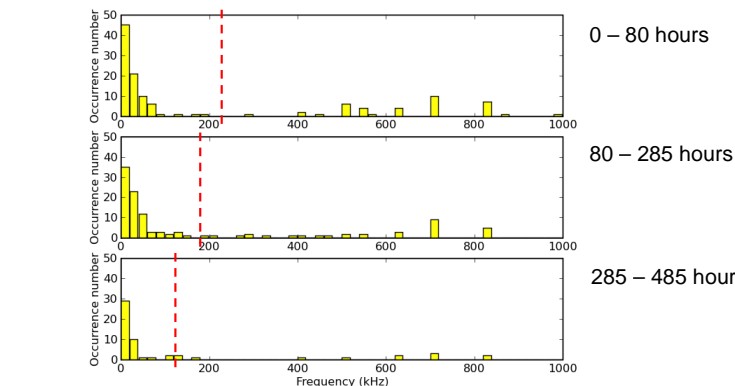

20

**Figure 8: Histograms representing successive statistical distribution of the AE signal frequencies with bins 20 kHz wide starting at 5 kHz. The dashed lines represent the mean frequency (224, 184 and 132 kHz values, respectively).**

### 5 Conclusion and prospectives

Structural monitoring based on the AE technique allowed to point out active microcracks in an arched structure located in

25    the Castle of Racconigi. Thus, the 3D localization of the ongoing damage process will result in more cost and time savings in case of future maintenance and intervention programs. Furthermore, a preliminary investigation of arch's critical state indicators was carried out using the natural time analysis applied to the AE time series. The obtained results apparently




reveal the possibility of capturing the transition of this structural element to a critical state through the analysis of natural time statistical parameters, such as the variance $\kappa_1$ and the entropy $S$.

On the other hands, the experimental evidence supports the hypothesis that a relevant part of AE activity emerging from the monitored element may be induced by evolving micro-seismicity falling into the preparation zone of a well identifiable

earthquake according to the Dobrovolsky criterion. Indeed, the relatively small number of inner AE sources localized into the structure, compared to the total amount of recorded AE events, is compatible with the existence of a scattered source, i.e., the crustal trembling.

Finally, AE structural monitoring potentially provides twofold information in seismic areas: one concerning the structural damage and the other concerning the microseismic activity, propagating across the ground-building foundation interface, for

which the building foundation represents a sort of extended underground probe (Gregori et al., 2004; Gregori et al., 2005; Carpinteri et al., 2007). In this sense, structural monitoring in seismic areas could be usefully coupled with investigations of the local earthquake precursors (Niccolini et al., 2015; Lacidogna et al., 2015b).

**Acknowledgements**

The previous Director of the Polo Museale del Piemonte del Real Castello e Parco di Racconigi Arch. Giuse Scalva and the

current Director Arch. Riccardo Vitale are gratefully acknowledged for their assistance in the definition of the site and during the phases of the monitoring.

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
