# Peer review of "Frequency signal and natural time analyses from acoustic emission monitoring of an arched structure in the Racconigi Castle"

_Natural Hazards and Earth System Sciences, 2016_

## Referee Comment (RC1) · Anonymous Referee #1 · 14 Feb 2017

**REF:** Manuscript : nhess-2016-374
   Title : Structural and Seismic Monitoring of a Monumental Building: the Case Study of the Royal Castle of Racconigi
   Author : Gianni Niccolini, Amedeo Manuello, Elena Marchis, Alberto Carpinteri

The authors' aim is to check the potential of a novel approach for detecting entrance to "critical conditions" that is based on the Natural Time concept. In this direction data recorded with the aid of the "Acoustic Emissions" technique are used.

In terms of a general overview:
- The content of the manuscript falls within the scope of "*Natural Hazards and Earth System Sciences*".
- The title of the manuscript does not properly reflect the content of the manuscript.
- The structure of the text should be reconsidered as it is described in the next section of this review.
- The conclusions drawn could be of interesting for the community of scientists working in the direction of detecting pre-failure indicators.
- The Reference list covers the field according to a satisfactory manner.

In terms of a detailed review, there are some critical points that, according to my opinion, should be taken into account by the authors. The revised manuscript must then be re-submitted for review. These points are described in the following paragraphs:

1. **The structure of the manuscript:**
   To my opinion the length of Section 2 is disproportionally long. The architectural and historical details are of minor importance (although could be of interest for the readers of journal oriented to Cultural Heritage themes) and the content of the specific section is rather irrelevant to the main target of the manuscript. The section should be drastically shortened, most of its figures must be removed, and then it could become part of the Introduction. On the contrary Section 4 should be analysed according to a much more detailed manner. Quite a few details concerning the way the raw data were processed are missing.

2. **The title of the manuscript:**
   The title of the contribution is somehow misleading or overambitious. The authors did not monitor a Monumental Building but rather a specific structural element of the building. Moreover the technique (Acoustic Emission) used to monitor the element should be somehow reflected in the title and the same is true for the analysis technique (Natural time).

3. **Assumptions adopted must be justified:**
   This is, for example, the case of the statement "*In particular, the increased AE rate marked by a vertical dashed line in the top diagram of Fig. 5 can be regarded as a signature of unstable damage accumulation*" (p.7, lines 5-7). I am not convinced that **any increase of the AE rate** is a sign of **unstable** damage accumulation.

   Along the same line the authors should justify their choice for "*… partitioning the time window preceding the considered seismic event into three sub-intervals (0-80h; 80h-285h; 285h-485h) roughly containing the same number of AE events*" (p.11, lines 6-9). What is the criterion for dividing the overall time interval into three sub-intervals and why the specific ones were chosen?

**4. Some qualitative statements and conclusions should be quantified:**
For example it is stated that "*…a progressive reduction of the highest frequencies, i.e., between 400 and 800 kHz, is observable as the seismic event was approached*" (p. 11, lines 9-10). What is the magnitude of this reduction?  Is this reduction significant and on what basis of comparison?

**5. Some typing errors must be corrected and some sentences should be rephrased.**
Although from a linguistic point of view the manuscript is very well written, some points must be considered, as follows:
**5.1** p.6, lines1-2. The sentence should be rephrased.
**5.2** p.8, line 17. The sentence contains a duplication of words.
**5.3** p.8, lines 22-26. This is a very long sentence and must be rephrased.
**5.4** p.12, line 1: "*On the other hands*" must be written "On the other hand".

Taking into account the as above thoughts, I believe that the manuscript could be accepted for publication, assuming that the authors would properly revise their manuscript according to the above mentioned  comments.

In this context my suggestion at this stage of the review process is:

**Revision and re-review.**

---

## Referee Comment (RC2) · Anonymous Referee #2 · 15 Feb 2017

In this manuscript (ms) the authors check the potential of a novel approach to identify the approach to a critical point on the basis of natural time analysis by employing acoustic emissions data.

**In general:**

This ms falls within the scope of "Natural Hazards and Earth System Sciences".

The title of the ms should be improved.

The conclusions drawn are very interesting.

The Reference list needs completion.

**More specifically:**

1. Concerning the structure of the ms, Section 2 should be shortened so that to become part of the Introduction. Furthermore, the authors should give more details in Section 4 on the way the raw data have been processed.

2. The title of the manuscript should be improved in order to become more precise. A tentative example may be: "Natural time analysis of acoustic emission from the Royal Castle of Racconigi"

3. Concerning the list of References:

In line 6 of p.8, Ref. [1] should be added to [33-35].

In line 18 of p.8, Ref. [2] should be inserted, because it is the one that gives the justification why $\kappa_1$ converges to 0.070 when the system enters the critical state.

In line 19 of p.8, just after the phrase "Two criteria ... to critical state:" the relevant Ref. [3] should be added.

In short, my suggestion in this stage of the review process is that the ms should be revised along the lines explained above.

References:

1. P. A. Varotsos, N. V. Sarlis, E. S. Skordas, and M. S. Lazaridou, Seismic Electric Signals: An additional fact showing their physical interconnection with seismicity, Tectonophysics 589 (2013) 116–125.

2. P. Varotsos, N.V. Sarlis, E.S. Skordas, S. Uyeda, and M. Kamogawa, Natural time analysis of critical phenomena. Proc. Natl. Acad. Sci USA 108 (2011), 11361-11364.

3. P.A. Varotsos, N.V. Sarlis, E.S. Skordas and M.S. Lazaridou, Fluctuations, under time reversal, of the natural time and the entropy distinguish similar looking electric signals of different dynamics, J. Appl. Phys. 103 (2008), 014906

---

## Author Comment (AC1) · 24 Feb 2017

We would like to thank the Referee for his precious and valuable suggestions. The answers to the referees points are presented here below. In the attached pdf file, we have included the revised copy of the manuscript with all suggested changes highlighted with a yellow background.

Manuscript ID: nhess-2016-374

Authors: Gianni Niccolini, Alberto Carpinteri, Amedeo Manuello, Elena Marchis

Title: "Structural and Seismic Monitoring of a Monumental Building: the Case Study of the Royal Castle of Racconigi"

1. STRUCTURE OF THE MANUSCRIPT Referee comment: To my opinion the length of Section 2 is disproportionly long [. . .]. The section should be drastically shortened, most of its figures must be removed, and then it could become part of the Introduction. On the contrary Section 4 should be analyzed according to a much more detailed manner. Quite a few details concerning the way the raw data were processed are missing. Authors' reply: Section containing architectural and historical details has been suppressed. Relevant information concerning the historical building and the structural element under monitoring became part of the Introduction, and we have removed figure 1 from the manuscript. On the other hand, we have given more experimental details, concerning data acquisition and data processing, in Section 2, now titled "Experimental results", and in Section 3, "Frequency and natural time analysis of AE time series and correlation with nearby seismicity". In Section 2, we have specified: • the type of the adopted transducers ("broad-band type, working in the range 10 kHz – 1 MHz"); • the accuracy of the arrival time of signals ("0.2 $\mu$s"); • the criterion used for determination of noise amplitude ("Before starting the monitoring, the background noise has been checked for a representative period of time, i.e. 24 hours, in order to determine the level of spurious signals: [. . .] 1.5 mV").

Further changes are reported in the following points.

2. THE TITLE OF THE MANUSCRIPT Referee comment: The title of the contribution is somehow misleading or overambitious. The authors did not monitor a Monumental Building but rather a specific structural element of the building. Moreover the technique (Acoustic Emission) used to monitor the element should be somehow reflected in the title and the same is true for the analysis technique (Natural time). Authors' reply: The title of the manuscript has been changed as follows: "Frequency and natural time analysis from acoustic emission monitoring of an arched structure in the Racconigi Castle".

3. ADOPTED ASSUMPTIONS THAT MUST BE JUSTIFIED Referee comment: This is, for example, the case of the statement "In particular, the increased AE rate marked

by a vertical dashed line in the top diagram of Fig. 5 can be regarded as a signature of unstable damage accumulation" (p.7, lines 5-7). I am not convinced that any increase of the AE rate is a sign of unstable damage accumulation. Authors' reply: We added the following statement in Section 2: "[. . .] Since all possible noisy signals in the frequency and amplitude range of measurement have been minimized, the burst of AE activity, marked by a vertical dashed line in the top diagram of Fig. 4, can be reasonably correlated with sudden increase in damage accumulation."

Referee comment: Along the same line the authors should justify their choice for ". . . partitioning the time window preceding the considered seismic event into three sub-intervals (0-80h; 80h-285h; 285h-485h) roughly containing the same number of AE events" (p.11, lines 6-9). What is the criterion for dividing the overall time interval into three sub-intervals and why the specific ones were chosen? Authors' reply: We added the following statement in Section 3: "We have chosen the following sub-intervals: (0h, 50h); (90h, 190h); (260h, 485h) characterized by different stages of the AE activity separated by quite long silent periods. The first interval, (0h-50h), contains a sudden increase in the AE rate, followed by two intervals, (90h, 190h) and (260h-485h), with smoother AE rates."

4. SOME QUALITATIVE STATEMENTS AND CONCLUSIONS SHOULD BE QUANTI-FIED Referee comment: For example it is stated that ". . .a progressive reduction of the highest frequencies, i.e., between 400 and 800 kHz, is observable as the seismic event was approached" (p. 11, lines 9-10). What is the magnitude of this reduction? Is this reduction significant and on what basis of comparison? Authors' reply: We added the following statement in Section 3: "The reduction is given in percentage terms, 30%, 22% and 16% of the total amount of signals for each distribution."

5. SOME TYPING ERRORS MUST BE CORRECTED AND SOME SENTENCES SHOULD BE REPHRASED Referee comment: Although from a linguistic point of view the manuscript is very well written, some points must be considered, as follows: 5.1 p.6, lines1-2. The sentence should be rephrased. 5.2 p.8, line 17. The sentence

contains a duplication of words. 5.3 p.8, lines 22-26. This is a very long sentence and must be rephrased. 5.4 p.12, line 1: "On the other hands" must be written "On the other hand". Authors' reply: 5.1 p.6, lines1-2. The sentence has been rephrased as follows: "Damage assessment in an arch of castle's thermal bath (Fig. 1) has been carried out by the AE technique, as a first step to plan possible restoration interventions. [. . .] The examined architectural element, currently supported by a steel frame structure, is a masonry arch with a span of 4 meters exhibiting a relevant crack pattern." 5.3 p.8, lines 22-26. The sentence has been rephrased as follows: "Here, the damage evolution of a structural element is investigated by analyzing the AE time series using two different methods and comparing the results. First, the evolution of variance $\kappa 1$ and entropy S of the natural-time transformed time series { $\chi$k } is studied, where the energy Qk associated with the AE event amplitude Ak is given by Qk = Ak1.5, similarly to seismology (Kanamori and Anderson, 1975; Turcotte, 1997). The second method used is the analysis of evolving AE signal frequencies over the monitoring time (Gregori et al., 2004; Gregori et al., 2005; Schiavi et al., 2011)."

Please also note the supplement to this comment:
http://www.nat-hazards-earth-syst-sci-discuss.net/nhess-2016-374/nhess-2016-374-AC1-supplement.pdf
* * *
[Figure]

**Supplement:**

**AUTHORS REPLY TO REVIEWER'S COMMENTS**

**Date: February, 24, 2017**

We would like to thank the Referee for his precious and valuable suggestions.
The answers to the referees points are presented here below. In the following, we have attached the revised copy of the manuscript with all suggested changes highlighted with a yellow background.

**Manuscript ID: nhess-2016-374**

**Authors: Gianni Niccolini, Alberto Carpinteri, Amedeo Manuello, Elena Marchis**

**Title: "Structural and Seismic Monitoring of a Monumental Building: the Case Study of the Royal Castle of Racconigi"**

**1. STRUCTURE OF THE MANUSCRIPT**

**Referee comment:**
To my opinion the length of Section 2 is disproportionally long […]. The section should be drastically shortened, most of its figures must be removed, and then it could become part of the Introduction. On the contrary Section 4 should be analyzed according to a much more detailed manner. Quite a few details concerning the way the raw data were processed are missing.

**Authors' reply:**
Section containing architectural and historical details has been suppressed. Relevant information concerning the historical building and the structural element under monitoring became part of the Introduction, and we have removed figure 1 from the manuscript. On the other hand, we have given more experimental details, concerning data acquisition and data processing, in Section 2, now titled "Experimental results", and in Section 3, "Frequency and natural time analysis of AE time series and correlation with nearby seismicity".
In Section 2, we have specified:

- the type of the adopted transducers ("*broad-band type, working in the range 10 kHz – 1 MHz*");
- the accuracy of the arrival time of signals ("*0.2 μs*");
- the criterion used for determination of noise amplitude ("*Before starting the monitoring, the background noise has been checked for a representative period of time, i.e. 24 hours, in order to determine the level of spurious signals: […] 1.5 mV*").

Further changes are reported in the following points.

**2. THE TITLE OF THE MANUSCRIPT**

**Referee comment:**
The title of the contribution is somehow misleading or overambitious. The authors did not monitor a Monumental Building but rather a specific structural element of the building. Moreover the technique (Acoustic Emission) used to monitor the element should be somehow reflected in the title and the same is true for the analysis technique (Natural time).

**Authors' reply:**
The title of the manuscript has been changed as follows: *"Frequency and natural time analysis from acoustic emission monitoring of an arched structure in the Racconigi Castle"*.

**3. ADOPTED ASSUMPTIONS THAT MUST BE JUSTIFIED**
**Referee comment:**
This is, for example, the case of the statement *"In particular, the increased AE rate marked by a vertical dashed line in the top diagram of Fig. 5 can be regarded as a signature of unstable damage accumulation"* (p.7, lines 5-7). I am not convinced that **any increase of the AE rate** is a sign of **unstable** damage accumulation.
**Authors' reply:**
We added the following statement in Section 2:
*"[…] Since all possible noisy signals in the frequency and amplitude range of measurement have been minimized, the burst of AE activity, marked by a vertical dashed line in the top diagram of Fig. 4, can be reasonably correlated with sudden increase in damage accumulation."*

**Referee comment:**
Along the same line the authors should justify their choice for *"… partitioning the time window preceding the considered seismic event into three sub-intervals (0-80h; 80h-285h; 285h-485h) roughly containing the same number of AE events"* (p.11, lines 6-9). What is the criterion for dividing the overall time interval into three sub-intervals and why the specific ones were chosen?
**Authors' reply:**
We added the following statement in Section 3: *"We have chosen the following sub-intervals: (0h, 50h); (90h, 190h); (260h, 485h) characterized by different stages of the AE activity separated by quite long silent periods. The first interval, (0h-50h), contains a sudden increase in the AE rate, followed by two intervals, (90h, 190h) and (260h-485h), with smoother AE rates."*

**4. SOME QUALITATIVE STATEMENTS AND CONCLUSIONS SHOULD BE QUANTIFIED**
**Referee comment:**
For example it is stated that *"…a progressive reduction of the highest frequencies, i.e., between 400 and 800 kHz, is observable as the seismic event was approached"* (p. 11, lines 9-10). What is the magnitude of this reduction? Is this reduction significant and on what basis of comparison?
**Authors' reply:**
We added the following statement in Section 3: *"The reduction is given in percentage terms, 30%, 22% and 16% of the total amount of signals for each distribution."*

**5. SOME TYPING ERRORS MUST BE CORRECTED AND SOME SENTENCES SHOULD BE REPHRASED**
**Referee comment:**
Although from a linguistic point of view the manuscript is very well written, some points must be considered, as follows:
**5.1** p.6, lines1-2. The sentence should be rephrased.
**5.2** p.8, line 17. The sentence contains a duplication of words.
**5.3** p.8, lines 22-26. This is a very long sentence and must be rephrased.
**5.4** p.12, line 1: *"On the other hands"* must be written "On the other hand".
**Authors' reply:**
**5.1** p.6, lines1-2. The sentence has been rephrased as follows:

[revised manuscript text omitted]

---

## Author Comment (AC2) · 24 Feb 2017

We would like to thank the Referee for his precious and valuable suggestions. The answers to the referees points are presented here below. In the attached pdf file, we have included the revised copy of the manuscript with all suggested changes highlighted with a yellow background.

Manuscript ID: nhess-2016-374

Authors: Gianni Niccolini, Alberto Carpinteri, Amedeo Manuello, Elena Marchis

Title: "Structural and Seismic Monitoring of a Monumental Building: the Case Study of the Royal Castle of Racconigi"

Referee comment: Concerning the structure of the ms, Section 2 should be shortened so that to become part of the Introduction. Furthermore, the authors should give more details in Section 4 on the way the raw data have been processed. Authors' reply: Section containing architectural and historical details has been suppressed. Relevant information concerning the historical building and the structural element under monitoring became part of the Introduction, and we have removed figure 1 from the manuscript. On the other hand, we have given more experimental details, concerning data acquisition and data processing, in Section 2, now titled "Experimental results", and in Section 3, "Frequency and natural time analysis of AE time series and correlation with nearby seismicity".

In Section 2, we have specified: • the type of the adopted transducers ("broad-band type, working in the range 10 kHz – 1 MHz"); • the accuracy of the arrival time of signals ("0.2 $\mu$s"); • the criterion used for determination of noise amplitude ("Before starting the monitoring, the background noise has been checked for a representative period of time, i.e. 24 hours, in order to determine the level of spurious signals: [. . .] 1.5 mV"). • we added the following statement: "[. . .] Since all possible noisy signals in the frequency and amplitude range of measurement have been minimized, the burst of AE activity, marked by a vertical dashed line in the top diagram of Fig. 4, can be reasonably correlated with sudden increase in damage accumulation."

We added the following statements in Section 3: • "We have chosen the following sub-intervals: (0h, 50h); (90h, 190h); (260h, 485h) characterized by different stages of the AE activity separated by quite long silent periods. The first interval, (0h-50h), contains a sudden increase in the AE rate, followed by two intervals, (90h, 190h) and (260h-485h), with smoother AE rates."

• "We have chosen the following sub-intervals: (0h, 50h); (90h, 190h); (260h, 485h) characterized by different stages of the AE activity separated by quite long silent periods. The first interval, (0h-50h), contains a sudden increase in the AE rate, followed by two intervals, (90h, 190h) and (260h-485h), with smoother AE rates."

• "The reduction is given in percentage terms, 30%, 22% and 16% of the total amount of signals for each distribution."

Two sentences have been rephrased as follows: • p.6, lines1-2 "Damage assessment in an arch of castle's thermal bath (Fig. 1) has been carried out by the AE technique, as a first step to plan possible restoration interventions. [. . .] The examined architectural element, currently supported by a steel frame structure, is a masonry arch with a span of 4 meters exhibiting a relevant crack pattern." • p.8, lines 22-26. The sentence has been rephrased as follows: "Here, the damage evolution of a structural element is investigated by analyzing the AE time series using two different methods and comparing the results. First, the evolution of variance $\kappa 1$ and entropy S of the natural-time transformed time series { $\chi$k } is studied, where the energy Qk associated with the AE event amplitude Ak is given by Qk = Ak1.5, similarly to seismology (Kanamori and Anderson, 1975; Turcotte, 1997). The second method used is the analysis of evolving AE signal frequencies over the monitoring time (Gregori et al., 2004; Gregori et al., 2005; Schiavi et al., 2011)."

Referee comment: The title of the manuscript should be improved in order to become more precise. A tentative example may be: "Natural time analysis of acoustic emission from the Royal Castle of Racconigi" Authors' reply: The title of the manuscript has been changed as follows: "Frequency and natural time analysis from acoustic emission monitoring of an arched structure in the Racconigi Castle".

Referee comment: Concerning the list of References: In line 6 of p.8, Ref. [1] should be added to [33-35]. In line 18 of p.8, Ref. [2] should be inserted, because it is the one that gives the justification why $\kappa 1$ converges to 0.070 when the system enters the critical state. In line 19 of p.8, just after the phrase "Two criteria ... to critical state:" the relevant Ref. [3] should be added. In short, my suggestion in this stage of the review process is that the ms should be revised along the lines explained above. References: 1. P. A. Varotsos, N. V. Sarlis, E. S. Skordas, and M. S. Lazaridou, Seismic Electric Signals: An additional fact showing their physical interconnection with seismicity,

**NHESSD**

Tectonophysics 589 (2013) 116–125. 2. P. Varotsos, N.V. Sarlis, E.S. Skordas, S. Uyeda, and M. Kamogawa, Natural time analysis of critical phenomena. Proc. Natl. Acad. Sci USA 108 (2011), 11361-11364. 3. P.A. Varotsos, N.V. Sarlis, E.S. Skordas and M.S. Lazaridou, Fluctuations, under time reversal, of the natural time and the entropy distinguish similar looking electric signals of different dynamics, J. Appl. Phys. 103 (2008), 014906 Authors' reply: We added the references according to referee's suggestions.

Please also note the supplement to this comment:
http://www.nat-hazards-earth-syst-sci-discuss.net/nhess-2016-374/nhess-2016-374-AC2-supplement.pdf

**Supplement:**

**AUTHORS REPLY TO REVIEWER'S COMMENTS**

**Date: February, 24, 2017**

We would like to thank the Referee for his precious and valuable suggestions.
The answers to the referees points are presented here below. In the following, we have attached the revised copy of the manuscript with all suggested changes highlighted with a yellow background.

**Manuscript ID: nhess-2016-374**

**Authors: Gianni Niccolini, Alberto Carpinteri, Amedeo Manuello, Elena Marchis**

**Title: "Structural and Seismic Monitoring of a Monumental Building: the Case Study of the Royal Castle of Racconigi"**

**Referee comment:**
Concerning the structure of the ms, Section 2 should be shortened so that to become part of the Introduction. Furthermore, the authors should give more details in Section 4 on the way the raw data have been processed.

**Authors' reply:**
Section containing architectural and historical details has been suppressed. Relevant information concerning the historical building and the structural element under monitoring became part of the Introduction, and we have removed figure 1 from the manuscript. On the other hand, we have given more experimental details, concerning data acquisition and data processing, in Section 2, now titled "Experimental results", and in Section 3, "Frequency and natural time analysis of AE time series and correlation with nearby seismicity".

In Section 2, we have specified:
- the type of the adopted transducers ("*broad-band type, working in the range 10 kHz – 1 MHz*");
- the accuracy of the arrival time of signals ("*0.2 μs*");
- the criterion used for determination of noise amplitude ("*Before starting the monitoring, the background noise has been checked for a representative period of time, i.e. 24 hours, in order to determine the level of spurious signals: [...] 1.5 mV*").
- we added the following statement:
  "*[...] Since all possible noisy signals in the frequency and amplitude range of measurement have been minimized, the burst of AE activity, marked by a vertical dashed line in the top diagram of Fig. 4, can be reasonably correlated with sudden increase in damage accumulation.*"

We added the following statements in Section 3:
- "*We have chosen the following sub-intervals: (0h, 50h); (90h, 190h); (260h, 485h) characterized by different stages of the AE activity separated by quite long silent periods. The first interval, (0h-50h), contains a sudden increase in the AE rate, followed by two intervals, (90h, 190h) and (260h-485h), with smoother AE rates.*"

- *"We have chosen the following sub-intervals: (0h, 50h); (90h, 190h); (260h, 485h) characterized by different stages of the AE activity separated by quite long silent periods. The first interval, (0h-50h), contains a sudden increase in the AE rate, followed by two intervals, (90h, 190h) and (260h-485h), with smoother AE rates."*

- *"The reduction is given in percentage terms, 30%, 22% and 16% of the total amount of signals for each distribution."*

Two sentences have been rephrased as follows:
- p.6, lines1-2 *"Damage assessment in an arch of castle's thermal bath (Fig. 1) has been carried out by the AE technique, as a first step to plan possible restoration interventions. […] The examined architectural element, currently supported by a steel frame structure, is a masonry arch with a span of 4 meters exhibiting a relevant crack pattern."*
- p.8, lines 22-26. The sentence has been rephrased as follows:
*"Here, the damage evolution of a structural element is investigated by analyzing the AE time series using two different methods and comparing the results. First, the evolution of variance $\kappa_1$ and entropy S of the natural-time transformed time series { $\chi_k$ } is studied, where the energy $Q_k$ associated with the AE event amplitude $A_k$ is given by $Q_k = A_k^{1.5}$, similarly to seismology (Kanamori and Anderson, 1975; Turcotte, 1997). The second method used is the analysis of evolving AE signal frequencies over the monitoring time (Gregori et al., 2004; Gregori et al., 2005; Schiavi et al., 2011)."*

**Referee comment:**
The title of the manuscript should be improved in order to become more precise. A tentative example may be: "Natural time analysis of acoustic emission from the Royal Castle of Racconigi"
**Authors' reply:**
The title of the manuscript has been changed as follows: *"Frequency and natural time analysis from acoustic emission monitoring of an arched structure in the Racconigi Castle"*.

**Referee comment:**
Concerning the list of References:
In line 6 of p.8, Ref. [1] should be added to [33-35].
In line 18 of p.8, Ref. [2] should be inserted, because it is the one that gives the justification why $\kappa_1$ converges to 0.070 when the system enters the critical state.
In line 19 of p.8, just after the phrase "Two criteria ... to critical state:" the relevant Ref. [3] should be added.
In short, my suggestion in this stage of the review process is that the ms should be revised along the lines explained above.

**Authors' reply:**
*We added the references according to referee's suggestions.*

[revised manuscript text omitted]

---

## Author Response (AR2)

**Manuscript ID: nhess-2016-374**

**Authors: Gianni Niccolini, Alberto Carpinteri, Amedeo Manuello, Elena Marchis**

**Dear Professor Vallianatos,**

**We would like to thank you and the referees again for your precious and valuable suggestions. Accordingly, we modified the title which now is:**

**"Signal frequency distribution and natural time analyses from acoustic emission monitoring of an arched structure in the Racconigi Castle"**

**Best regards,**

**Gianni Niccolini**